# Massive image-based single-cell profiling reveals high levels of circulating platelet aggregates in patients with COVID-19

Masako Nishikawa[1,13], Hiroshi Kanno [2,13], Yuqi Zhou[2,13], Ting-Hui Xiao [2✉], Takuma Suzuki[3], Yuma Ibayashi[2], Jeffrey Harmon[2], Shigekazu Takizawa[2], Kotaro Hiramatsu [2,4], Nao Nitta [5], Risako Kameyama[2], Walker Peterson[2], Jun Takiguchi[1], Mohammad Shifat-E-Rabbi[6], Yan Zhuang[7], Xuwang Yin[7], Abu Hasnat Mohammad Rubaiyat[7], Yunjie Deng[2], Hongqian Zhang[2], Shigeki Miyata[8], Gustavo K. Rohde[6,7], Wataru Iwasaki [3,9,10], Yutaka Yatomi [1✉] & Keisuke Goda [2,11,12✉]

A characteristic clinical feature of COVID-19 is the frequent incidence of microvascular thrombosis. In fact, COVID-19 autopsy reports have shown widespread thrombotic micro-angiopathy characterized by extensive diffuse microthrombi within peripheral capillaries and arterioles in lungs, hearts, and other organs, resulting in multiorgan failure. However, the underlying process of COVID-19-associated microvascular thrombosis remains elusive due to the lack of tools to statistically examine platelet aggregation (i.e., the initiation of micro-thrombus formation) in detail. Here we report the landscape of circulating platelet aggregates in COVID-19 obtained by massive single-cell image-based profiling and temporal monitoring of the blood of COVID-19 patients ($n = 110$). Surprisingly, our analysis of the big image data shows the anomalous presence of excessive platelet aggregates in nearly 90% of all COVID-19 patients. Furthermore, results indicate strong links between the concentration of platelet aggregates and the severity, mortality, respiratory condition, and vascular endothelial dysfunction level of COVID-19 patients.

[1] Department of Clinical Laboratory Medicine, Graduate School of Medicine, The University of Tokyo, Tokyo 113-0033, Japan. [2] Department of Chemistry, The University of Tokyo, Tokyo 113-0033, Japan. [3] Department of Computational Biology and Medical Sciences, The University of Tokyo, Chiba 277-8562, Japan. [4] Research Center for Spectrochemistry, The University of Tokyo, Tokyo 113-0033, Japan. [5] CYBO, Inc, Tokyo 135-0064, Japan. [6] Department of Biomedical Engineering, University of Virginia, Virginia 22908, USA. [7] Department of Electrical and Computer Engineering, University of Virginia, Virginia 22908, USA. [8] Research and Development Department, Central Blood Institute, Japanese Red Cross Society, Tokyo 135-8521, Japan. [9] Department of Biological Sciences, The University of Tokyo, Tokyo 113-0033, Japan. [10] Department of Integrated Biosciences, The University of Tokyo, Chiba 277-8562, Japan. [11] Institute of Technological Sciences, Wuhan University, 430072 Hubei, China. [12] Department of Bioengineering, University of California, Los Angeles, California 90095, USA. [13]These authors contributed equally: Masako Nishikawa, Hiroshi Kanno, Yuqi Zhou. ✉email: xiaoth@chem.s.u-tokyo.ac.jp; yatoyuta-tky@umin.ac.jp; goda@chem.s.u-tokyo.ac.jp

With the increasing number of global case reports since the beginning of the COVID-19 pandemic, it has become clear that COVID-19-associated microvascular thrombosis is one of the key factors for the severity and mortality of COVID-19[1–11]. In fact, earlier autopsy reports on patients who died with COVID-19 have shown widespread thrombotic microangiopathy (TMA) characterized by extensive diffuse microthrombi present within peripheral capillaries and arterioles in lungs, hearts, and other organs, resulting in multi-organ failure[1,3,7–11]. This is aligned with respiratory failure due to severe diffuse alveolar damage being the primary cause of death in COVID-19[3,10]. It is also consistent with our current understanding of the pathophysiological mechanism of COVID-19 in which SARS-CoV-2's entry into host cells is mediated by the angiotensin-converting enzyme 2 (ACE2) receptor, which stimulates the renin-angiotensin-aldosterone system and causes vascular endothelial damage followed by vasculitis (i.e., the inflammation of blood vessels), resulting in the formation of microthrombi[2,6]. In response to a number of reports that anticoagulant therapy with heparin leads to better prognosis in COVID-19 patients[12], both domestic and international organizations have issued clinical practice guidelines recommending that all hospitalized COVID-19 patients should receive thromboprophylaxis (mainly heparin treatment) even without clear symptoms of thrombotic complications and understanding their efficacy[13,14]. Moreover, post-COVID-19 syndrome manifested by persistent and prolonged aftereffects has been reported to be linked to COVID-19-associated microvascular thrombosis[15,16].

Unfortunately, the underlying process of the incidence of COVID-19-associated microvascular thrombosis remains elusive. This is due to the lack of tools to statistically examine the detailed characteristics of platelet activity, or more specifically platelet aggregation (i.e., the initiation of thrombus formation)[17–20] in vivo. For example, optical microscopy[21] can directly probe platelet aggregation with high spatial resolution and has, in fact, visualized platelet hyperactivity in COVID-19 by identifying the presence of platelet aggregates (i.e., thrombus constituents) including macrothrombocytes[22]. However, visual inspection under the optical microscope is too slow and labor-intensive to analyze platelet aggregates in a statistically meaningful manner and can only provide qualitative results. Flow cytometry, on the other hand, enables statistical analysis of large populations of cells by measuring their physical and chemical properties via impedance, scattering, or fluorescence measurements in flow[23–27] and has been used to identify leukocyte hyperactivity in COVID-19 via fluorescent probes[28–30]. However, the lack of spatial resolution in flow cytometry prohibits accurate differentiation between single platelets and platelet aggregates[26,31,32] and hence cannot accurately characterize platelet activity. Likewise, light transmission aggregometry[33], although widely used in clinical laboratory testing, only evaluates the average agonist-induced aggregation properties of many platelets in vitro through ensemble measurements and cannot provide details of platelet aggregation such as the size distribution or single-to-aggregate population ratio. Moreover, D-dimer testing, another widely used diagnostic, is conventionally used to estimate the presence of thrombi by measuring the cross-linked fibrin monomers (called D-dimers) produced when thrombi are degraded by fibrinolysis[34]. D-dimer testing has been reported as useful for assessing the severity of COVID-19, as an elevated level of D-dimers (>1 μg/mL) at admission is associated with an increased risk of both required mechanical ventilation and death due to complications[1,3,9,12,35,36]. However, D-dimer testing does not probe platelet activity in vivo and is not sensitive to microvascular thrombosis including TMA (unless it becomes severe)[37]. Medical imaging techniques such as computed tomography (CT), magnetic resonance imaging (MRI), and ultrasonography can directly identify thrombi in the body, but do not have sufficient spatial resolution to recognize circulating platelet aggregates and microthrombi[38]. Finally, postmortem examination can directly identify microthrombi but is only applicable to dead patients. To date, statistical morphometric understanding of platelet aggregation has been inaccessible and hence overlooked, as optical microscopy (a high-content but low-throughput tool) has been the main method used to examine platelet aggregation in detail thus far[26,32].

In this Article, to better comprehend the transient process of microthrombus formation in COVID-19, we obtained the landscape of circulating platelet aggregates via massive single-cell image-based profiling and temporal monitoring of the blood of COVID-19 patients. As shown in Fig. 1a, our data-driven analysis method consists of (i) blood draw (1 mL) from COVID-19 patients (Supplementary Data 1); (ii) sample preparation by isolating platelets and platelet aggregates from the blood; (iii) high-throughput, blur-free, bright-field imaging of a large population (n = 25,000) of single platelets and platelet aggregates (including platelet-platelet aggregates and platelet-leukocyte aggregates[39]) in blood samples, by optical frequency-division-multiplexed (FDM) microscopy[32] on a hydrodynamic-focusing microfluidic chip[40,41] (Fig. 1b, c, Supplementary Fig. 1, see "optical frequency-division-multiplexed microscope" in the Methods section for details); and (iv) digital image processing and the application of various techniques for statistical platelet aggregate analysis. The FDM microscope's image acquisition of all platelet events (e.g., single platelets, platelet-platelet aggregates, and platelet-leukocyte aggregates) was triggered by detecting fluorescence signals from anti-CD61-PE-labeled platelets (see "optical frequency-division-multiplexed microscope" in the Methods section for details), which is advantageous over previous imaging flow cytometers[42,43] in that our method can only focus on platelet events while avoiding its throughput from being consumed by non-platelet events. Image acquisition and clinical laboratory testing were performed at a frequency of 3-5 times per week per hospitalized patient to observe temporal changes in the population and size distribution of platelet aggregates during his/her hospitalization (Supplementary Data 2). Figure 1d shows a library of typical bright-field images (67 × 67 pixels/image) of single platelets and platelet aggregates flowing at a high speed of 1 m/s, acquired within a field of view of 53.6 μm × 53.6 μm with a spatial resolution of 0.8 μm (Supplementary Fig. 2). As shown in the figure, the ability of our approach to directly image and resolve single platelets, platelet-platelet aggregates, platelet-leukocyte aggregates, and other unimportant components (e.g., residual red blood cells, leukocytes, and cell debris) made it possible to differentiate them with a much higher accuracy than conventional flow cytometry[23–26] (especially in discriminating between single platelets and small platelet aggregates). Importantly, this big image database of platelet aggregates enabled the previously unfeasible application of advanced computational tools[31,44] which provided unique and insightful analytical results, as we demonstrated here.

## Results

**Image acquisition and size distribution.** We acquired 25,000 bright-field images of single platelets and platelet aggregates in the blood of hospitalized patients (n = 110) who were clinically diagnosed with COVID-19 based on their reverse transcription-polymerase chain reaction (RT-PCR) test results (Fig. 2a and Supplementary Data 1). Negative control image data were also obtained from healthy subjects under the same sample preparation and image acquisition conditions on the same day to mitigate potential bias in the image data that may have come from

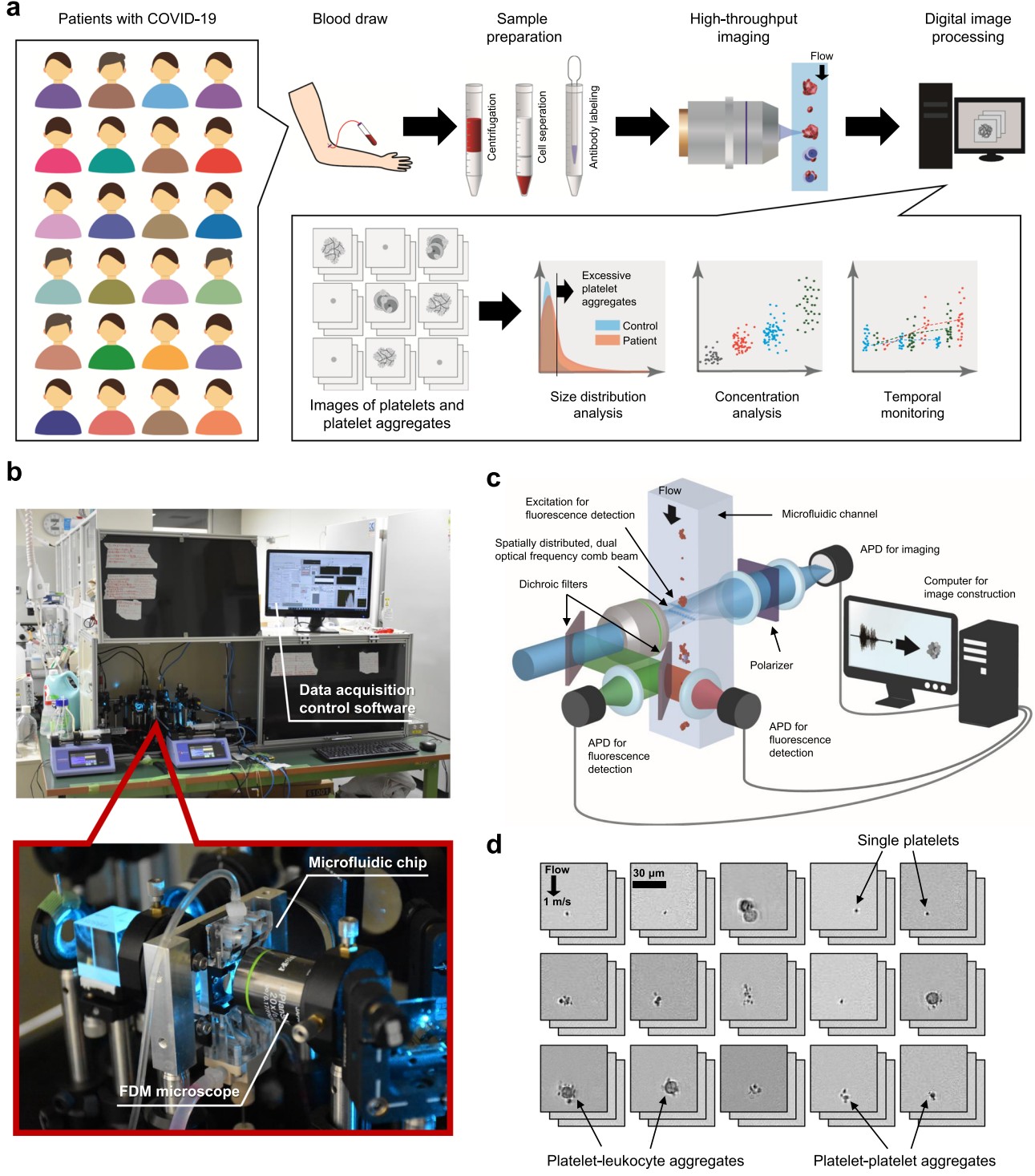

**Fig. 1 Massive image-based profiling of circulating platelets and platelet aggregates at single-cell resolution. a** Experimental workflow consisting of blood draw, sample preparation, high-throughput imaging, and digital image analysis. **b** Pictures of the FDM microscope on the microfluidic chip installed in the Department of Clinical Laboratory at the University of Tokyo Hospital. The inset shows an enlarged view of the microfluidic chip with the FDM microscope. **c** Schematic of the FDM microscope for high-throughput, blur-free, bright-field image acquisition. APD: avalanche photodetector. **d** Typical bright-field images of single platelets and platelet aggregates acquired by the FDM microscope. For information about data reproducibility, see "human subjects" and "statistical analysis" in the Methods section for details.

experimental variations in blood draw, sample preparation, optical alignment, and hydrodynamic focusing conditions and hence to maintain the state of platelet aggregation in vivo while minimizing the effect of aggregation in vitro (see "sample preparation" in the Methods section for details of the sample preparation protocol). Image acquisition was performed at a high throughput of 100–300 events per second (eps), where an event is defined as a single platelet or a platelet aggregate. Residual components such as red blood cells, leukocytes (excluding those contained in platelet-leukocyte aggregates), and cell debris were ignored and not detected as events. The throughput was chosen to avoid clogging the microchannel in the microfluidic chip,

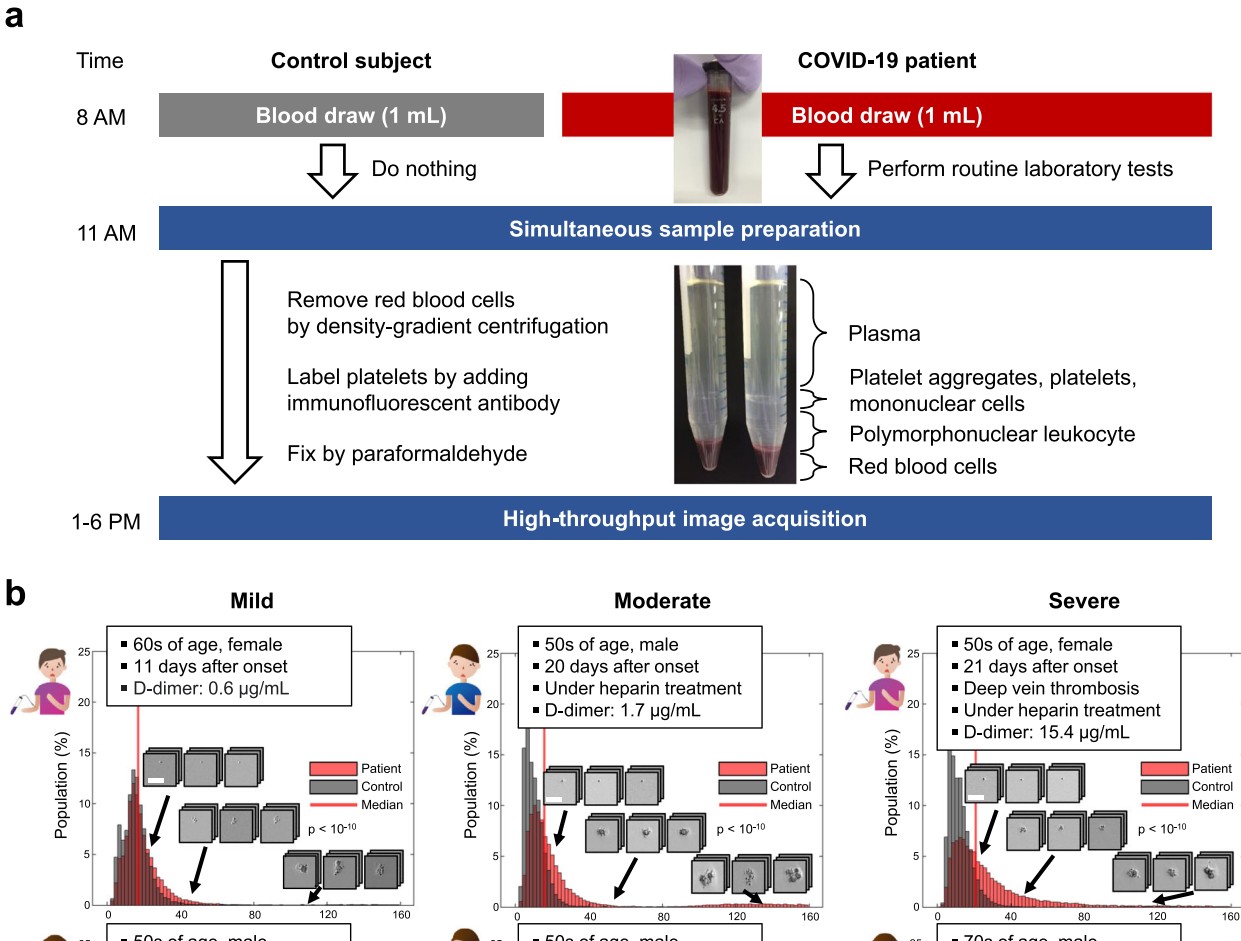

**Fig. 2 Circulating platelet aggregates in the blood of COVID-19 patients. a** Sample preparation protocol. Control data were also obtained under the same sample preparation and image acquisition conditions on the same day to mitigate potential bias in the image data that may have come from experimental variations in blood draw, sample preparation, optical alignment, and hydrodynamic focusing conditions. See "Sample preparation" in the Methods section for details of the protocol. **b** Size distribution histograms and typical images of single platelets and platelet aggregates in the blood of mild, moderate, and severe patients with COVID-19. Scale bar: 30 μm. The clinical information and D-dimer level of each patient are also shown in the insets. Control data are also shown as references. *P* values were obtained using the Wilcoxon rank-sum test (two-sided) and shown in the figure. Source data are provided as a Source Data file.

although the theoretical throughput of the FDM microscope was >10,000 eps.

Shown in Fig. 2b are size distribution histograms and typical images of single platelets and platelet aggregates identified in the blood of patients with COVID-19 (see the "human subjects" in the Methods section and the complete dataset in Source Data 1). The clinical information and D-dimer level of each patient are also shown in the figure insets. The negative control data are also shown in the figure as references. The objective area was defined by the area of a detected event (i.e., a single platelet or a single platelet aggregate) within each image. The COVID-19 patients were categorized into three groups: (i) the mild patient group: those requiring no oxygen therapy; (ii) the moderate patient group: those requiring oxygen therapy; and (iii) the severe patient group: those requiring mechanical ventilation or extracorporeal membrane oxygenation (ECMO) for respiratory support. Each patient's highest severity level during his/her hospitalization was used for the categorization. As shown in Fig. 2b, excessive platelet aggregates were also present in the blood of patients with low D-dimer levels (≤1 μg/mL), which suggests the presence of enhanced platelet activity by COVID-19, but before the potential onset of clinically evident thrombosis (i.e., the formation and degradation of microthrombi).

**Statistical analysis**. We performed a statistical analysis of the big image data of platelet aggregates to visualize trends and correlations. Fig. 3a compares the platelet aggregate concentrations of mild, moderate, and severe COVID-19 patients on the highest concentration day of each hospitalized patient. The concentration of platelet aggregates was defined by the ratio of the number of acquired images containing platelet aggregates to the total number of acquired images ($n = 25,000$) in each sample. Negative control data (shown as "control" in Fig. 3a–c) were provided by blood samples from healthy subjects ($n = 4$) on 67 different dates (see positive control data in Supplementary Fig. 3). Interestingly, the figure shows the anomalous presence of excessive platelet aggregates in as many as 87.3% of all COVID-19 patients, including those with D-dimer levels below the reference level of ≤1 μg/mL at the University of Tokyo Hospital (see "statistical analysis" in the Methods section for details and see the Discussion section for interpreting the concentration of platelet aggregates in the remaining 12.7% of patients). The excessive platelet aggregation identified in each individual sample is consistent with earlier reports on averaged platelet hyperactivity detected by platelet function tests and gene expression analysis[45–47]. The figure also indicates a strong link between the severity of patients and the concentration of platelet aggregates. Moreover, results show an increase in the concentrations of platelet aggregates between measurements taken on the first day after admission (Fig. 3b) and on the highest concentration day (Fig. 3a) (typically ~1 week after the first measurement day), in agreement with previous reports on COVID-19 patients whose condition worsened ~1 week after their initial symptoms appeared[1,5,6,35]. Likewise, Fig. 3c shows a strong correlation between the mortality of patients and the concentration of platelet aggregates. Fig. 3d–f show receiver operating characteristic (ROC) curves of the concentration of platelet aggregates with respect to the control corresponding to the data shown in Fig. a–c, respectively. The large value of the area under the curve (AUC) in each ROC curve indicates the excellent performance of the concentration of platelet aggregates in evaluating COVID-19-associated platelet activity.

Moreover, we investigated the influence of sex differences and leukocytes. Specifically, Fig. 4a shows a higher concentration of platelet aggregates in male patients ($n = 73$) than in female patients ($n = 37$) although their statistical significance is not very high, which is aligned with earlier reports that male patients are more prone to intensive treatment unit admission and death than female patients[48]. In addition, the ability of our method to highly resolve platelet aggregates and apply morphometric analysis to their images enabled identifying the presence of leukocytes. Fig. 4b, c show that the presence of leukocytes in platelet aggregates is linked with the severity and mortality of COVID-19, respectively, which is also consistent with previous reports on leukocyte hyperactivity in COVID-19[28–30,35]. Fig. 4d, e show ROC curves of the concentration of platelet aggregates with respect to the control corresponding to the data shown in Fig. 4b, c, respectively. The AUCs in these ROC curves indicate the excellent performance of the concentration of platelet aggregates containing leukocytes in evaluating COVID-19-associated platelet-leukocyte activity.

**Comparison with clinical laboratory and physical findings**. We compared the landscape of circulating platelet aggregates with conventional clinical laboratory and physical findings (Supplementary Data 1, Supplementary Data 3, see "clinical laboratory tests" in the Methods section for details). As shown in Fig. 5a, the leukocyte count (WBC), D-dimer level, coagulation factor VIII activity level (FVIII), von Willebrand factor activity level (VWF:RCo), thrombomodulin level (TM), and oxygen administration severity level (respiratory severity) were found to be statistically relevant with the concentration of platelet aggregates measured on the highest concentration day, as indicated by Spearman's rank correlation coefficients greater than 0.4 for all these parameters (Supplementary Fig. 4), whereas the other parameters such as the red blood cell count (RBC), C-reactive protein concentration (CRP), fibrinogen level (Fbg), creatinine concentration (Cre), and body temperature were weakly correlated or uncorrelated with it (Supplementary Fig. 5). Additionally, it should be mentioned that the correlation between the concentration of platelet aggregates and the SpO$_2$ (oxygen saturation level) level is reasonable since oxygen administration required to maintain the SpO$_2$ level in good condition reflects the disease state of patients. These strong correlations between the concentration of platelet aggregates and the WBC, FVIII, VWF:RCo, and TM levels indicate that the high concentration of platelet aggregates is linked with systemic microthrombus formation and fibrinolysis, which are expressed by the high D-dimer level, and with vascular endothelial damage (i.e., vasculitis), which is expressed by the high FVIII, VWF:RCo, and TM levels. These links are consistent with earlier reports on severe vascular endothelial damage in the lungs and widespread microthrombi in the alveolar capillaries of COVID-19 patients[3,10,49].

To further investigate significant explanatory factors for the concentration of platelet aggregates, we performed a multivariate regression analysis (SPSS software version 25, IBM) using the vascular endothelial disorder marker group (FVIII, TM, VWF:RCo), the coagulation/fibrinolysis marker group (D-dimer, FDP, TAT, PT-INR), and the group of other markers having strong correlations with the concentration of platelet aggregates on the highest concentration day (WBC, respiratory severity, PLT, LD, ALT, SpO$_2$, survival, sex) as potential explanatory factors (Supplementary Data 4). As a result, the analysis identified FVIII, PLT, WBC, PT-INR, and respiratory severity as significant predictors of the concentration of platelet aggregates on the highest concentration day. A further multivariate regression analysis using these parameters identified respiratory severity ($p = 0.009$) and FVIII ($p = 0.016$) as the most significant predictors of the concentration of platelet aggregates on the highest concentration day.

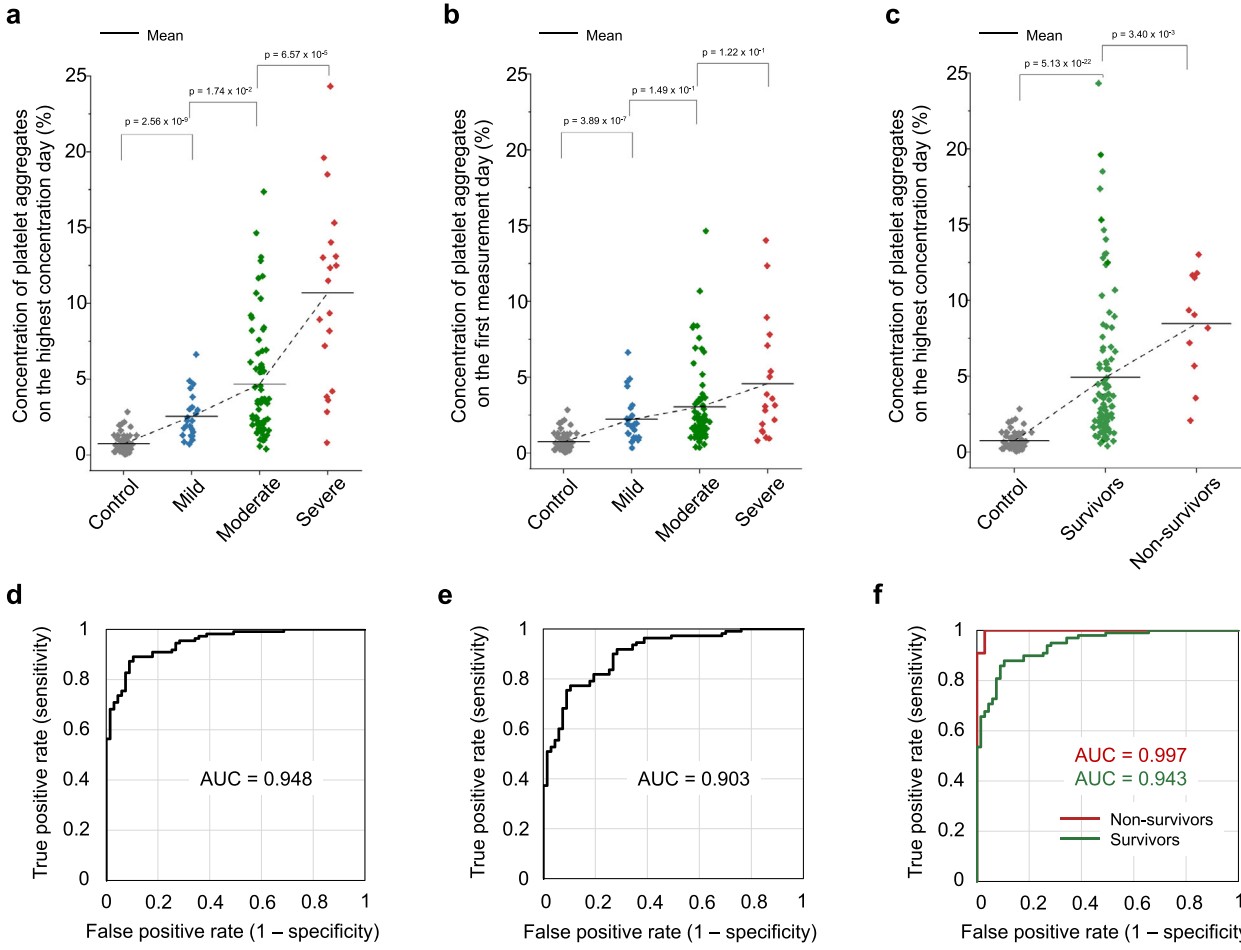

**Fig. 3 Comparison with the severity and mortality of COVID-19.** *P* values of every two adjacent classes were obtained using the Mann–Whitney *U*-test (two-sided) and shown in the figure. **a** Comparison of the platelet aggregate concentrations of mild (*n* = 23 biologically independent samples), moderate (*n* = 68 biologically independent samples), and severe (*n* = 19 biologically independent samples) patients with COVID-19, measured on the highest concentration day of each hospitalized patient. $p = 7.41 \times 10^{-24}$ with the two-sided Kruskal–Wallis test. For the definition of images containing platelet aggregates, see "statistical analysis" in the Methods section for details. **b** Comparison of the platelet aggregate concentrations of mild (*n* = 23 biologically independent samples), moderate (*n* = 68 biologically independent samples), and severe (*n* = 19 biologically independent samples) patients with COVID-19 on the first measurement day of each hospitalized patient. $p = 4.37 \times 10^{-18}$ with the two-sided Kruskal–Wallis test. **c** Comparison of the platelet aggregate concentrations of survivors (*n* = 99 biologically independent samples) and non-survivors (*n* = 11 biologically independent samples) from COVID-19, measured on the highest concentration day of each hospitalized patient. $p = 3.05 \times 10^{-23}$ with the two-sided Kruskal–Wallis test. **d, e, f,** ROC curves of the data shown in Figs. 3a–c, respectively. Source data are provided as a Source Data file.

Next, since earlier reports show the onset of COVID-19-associated thrombosis even with low D-dimer levels[4], we investigated patient cases with D-dimer levels below the reference level of ≤1 μg/mL, which constitutes 39.1% of all COVID-19 patients (n = 110). As shown in Fig. 5b, significant levels of platelet aggregate concentrations are evident. Also, as shown in Fig. 5c, the correlation between the severity of patients and the concentration of platelet aggregates also persists and is similar to the relation shown in Fig. 3a although the difference between the mild and moderate patients is not very large, which is reasonable because their medical conditions (both without mechanical ventilation or ECMO) are similar. The figure also shows that excessive platelet aggregates were found in as many as 76.7% of patients with a D-dimer level of ≤1 μg/mL (see "statistical analysis" in the Methods section for details). These results may suggest the presence of a non-negligible microvascular thrombotic risk that could not be detected by the D-dimer test. In other words, the D-dimer test mainly evaluates blood coagulation by measuring D-dimers that are produced either at the final stage of thrombus formation (i.e., cross-linking) or after thrombus degradation[34] and is known to be insensitive to microvascular thrombosis, including TMA[37], whereas our method characterizes the initiation of microthrombus formation and the degradation of microthrombi[17–20]. The latter method is, therefore, more sensitive with high resolution to platelet hyperactivity, which is a suggested mechanism of widespread microthrombus formation[2,45]. In addition, the fact that only one severe case with a low D-dimer level was identified as shown in Fig. 5c can be explained by recognizing that the D-dimer test is generally insensitive to TMA as mentioned above, but can detect it if the medical conditions of patients become severe[50]. In other words, the correlation between the D-dimer level and the concentration of platelet aggregates in Fig. 5a is valid for moderate-to-severe cases while not significant for mild-to-moderate cases. Finally, as shown in Fig. 5d, the large AUC value of the ROC curve of the concentration of platelet aggregates with low D-dimer levels of ≤1 μg/mL with respect to the control indicates our method's high performance in assessing the severity of COVID-19 even when the D-dimer level is low.

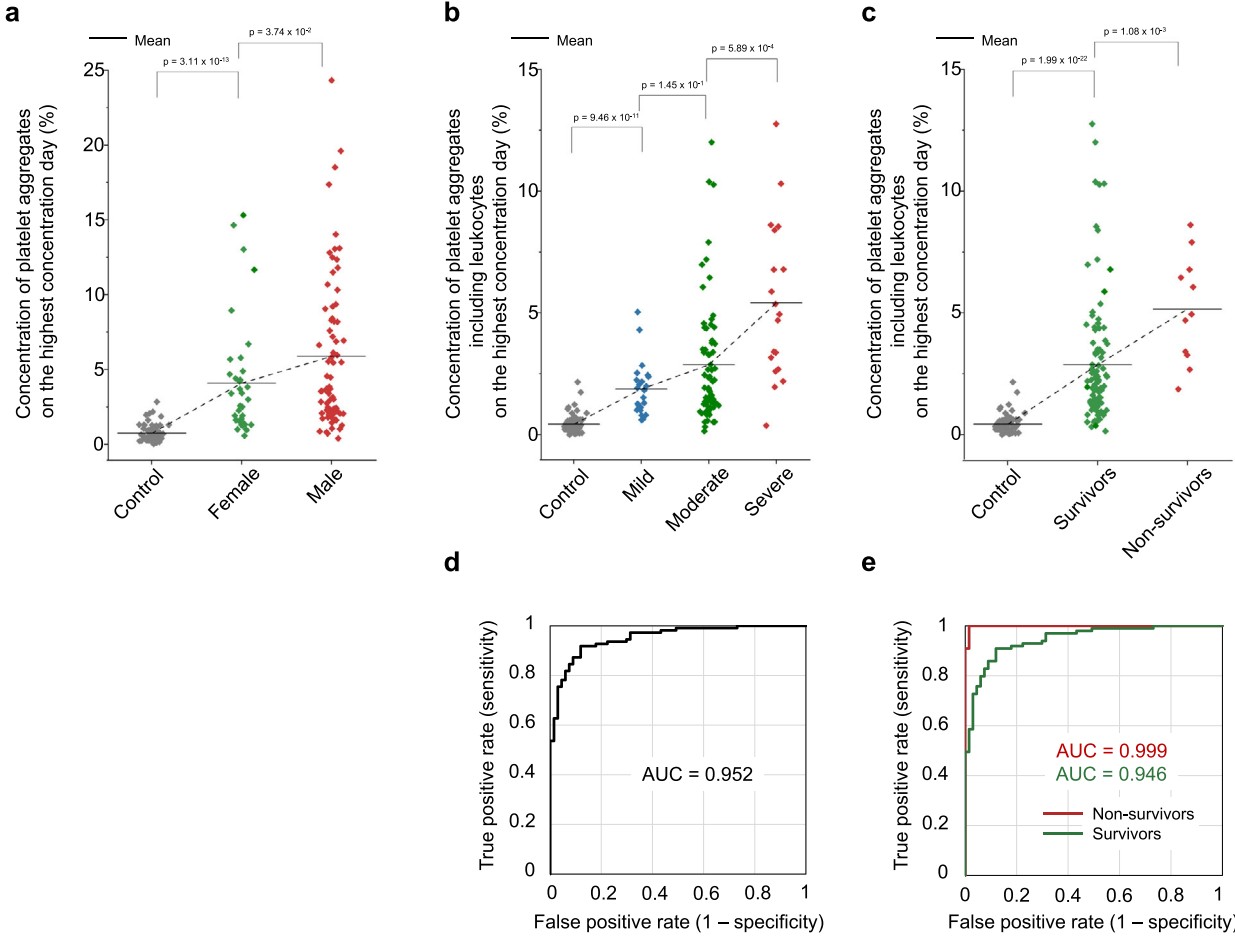

**Fig. 4 Influence of sex differences and leukocytes.** *P* values of every two adjacent classes were obtained using the Mann–Whitney *U*-test (two-sided) and shown in figures. **a** Comparison of the platelet aggregate concentrations of male (*n* = 73) and female (*n* = 37 biologically independent samples) patients, measured on the highest concentration day of each hospitalized patient. $p = 8.97 \times 10^{-23}$ with the two-sided Kruskal–Wallis test. **b** Comparison of the concentrations of platelet aggregates of mild (*n* = 23 biologically independent samples), moderate (*n* = 68 biologically independent samples), and severe (*n* = 19 biologically independent samples) patients with COVID-19 including leukocytes, measured on the highest concentration day of each hospitalized patient. $p = 1.88 \times 10^{-23}$ with the two-sided Kruskal–Wallis test. **c** Comparison of the concentrations of platelet aggregates of survivors (*n* = 99 biologically independent samples) and non-survivors (*n* = 11 biologically independent samples) from COVID-19, measured on the highest concentration day of each hospitalized patient. $p = 7.91 \times 10^{-24}$ with the two-sided Kruskal–Wallis test. **d**, **e** ROC curves of the data shown in Fig. 4b, c, respectively. Source data are provided as a Source Data file.

**Temporal monitoring.** Our temporal monitoring of the concentration of platelet aggregates in the blood of COVID-19 patients enabled us to probe the pathological conditions of COVID-19 patients, 76% of whom received anticoagulant therapy with heparin. Figure 6a shows the evolution of the platelet aggregate concentration of each patient group (mild, moderate, severe) after the onset of COVID-19 (see "human subjects" in the Methods section, the complete dataset in Source Data 1, and the measurement days in Supplementary Data 2). As mentioned above, each patient's highest severity level during his/her hospitalization was used for the severity categorization. Specifically, as shown in the figure, the platelet aggregate concentration of the mild patient group reached a peak (about 2.5%) in the first 9–12 days and then gradually decreased over a week, followed by the complete discharge of all the mild patients from the hospital after 16 days. Likewise, the platelet aggregate concentration of the moderate patient group reached a peak (about 5%) after the first 13–16 days and then gradually decreased over two weeks, followed by the complete discharge of all the moderate patients from the hospital after 28 days. On the other hand, the platelet aggregate concentration of the severe patient group initially increased to a peak (about 7%) in the first week and

then exhibited a plateau for three weeks, followed by death or transfer to a chronic hospital. It is important to note that all the patient groups showed a moderately high platelet aggregate concentration level (about 2.5%) in the first 3–4 days, but their concentrations started to differ in the next 3–4 days, such that each patient group underwent a different prognosis pattern while the timing of discharge from the hospital was consistent with the decreased concentration of platelet aggregates in all prognosis patterns. Also, as shown in the figure, patients who developed thrombosis ended up staying in the hospital longer than the mild and moderate patients and some of the severe patients. Moreover, as shown in Fig. 6b, the strong correlation between the respiratory severity level and the concentration of platelet aggregates in the first two periods (day 1–9, day 10–18) indicates that the concentration of platelet aggregates is a good indicator of the respiratory condition of COVID-19 patients. The twin-peak distribution of the platelet aggregate concentrations of patients in the last period (≥ day 19 after the onset of COVID-19) indicates that most patients either ended up recovering or remained hospitalized as characterized by the patients at levels 2 and 3 (those requiring oxygen administration) stepping down to level 1 (no oxygen administration) and

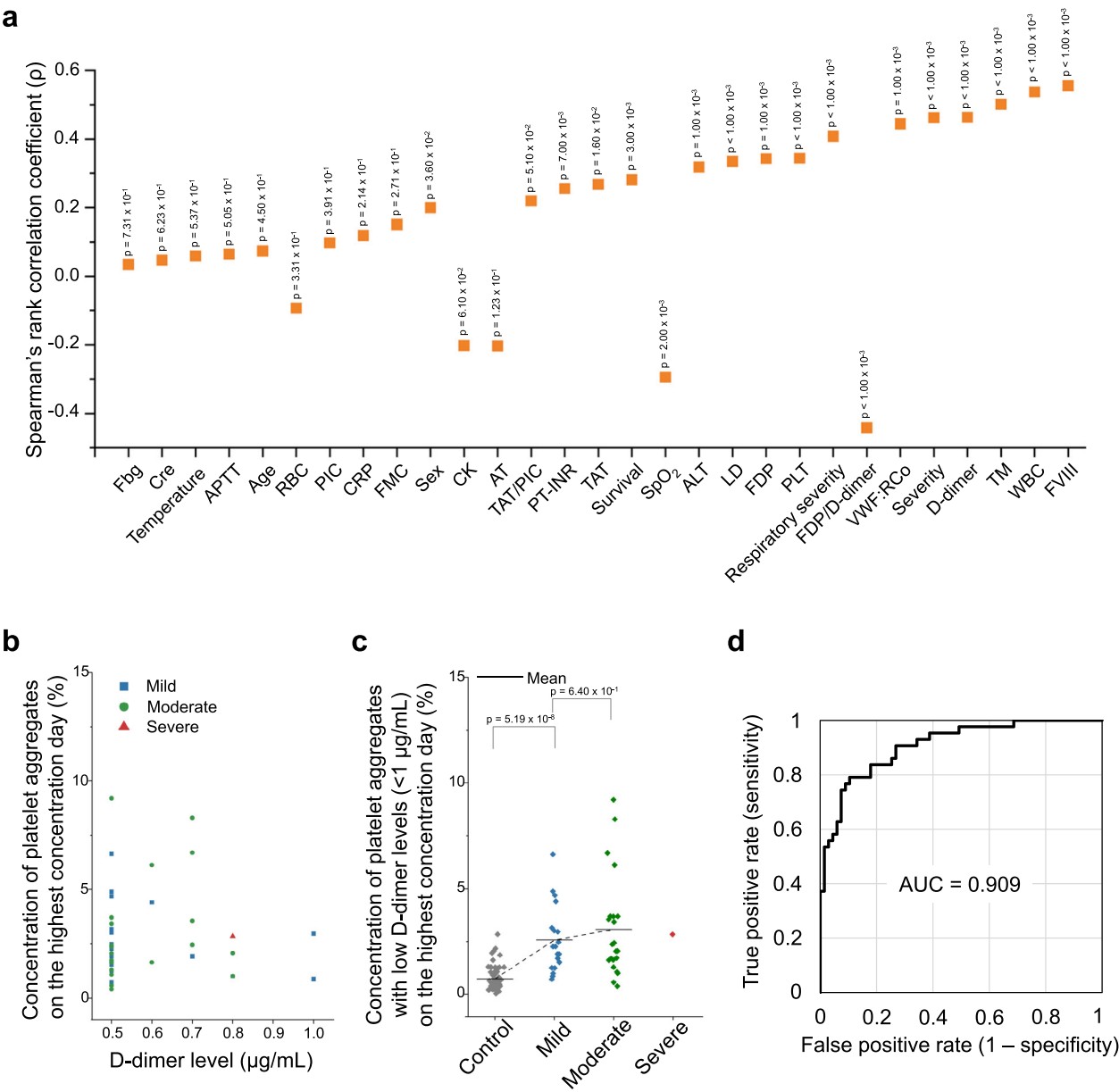

**Fig. 5 Comparison with clinical laboratory tests. a** Comparison of the platelet aggregate concentration measured on the highest concentration day with clinical laboratory and physical findings. WBC leukocyte count, RBC red cell count, PLT platelet count, APTT activated partial thromboplastin time, PIC plasma plasmin-α2-plasmin inhibitor complex, CRP C-reactive protein concentration, FMC fibrin monomer complex, CK creatinine kinase, AT antithrombin, PT-INR prothrombin time international normalized ratio, TAT thrombin antithrombin complex, $SpO_2$ oxygen saturation, ALT alanine transaminase concentration, LD lactate dehydrogenase concentration, FDP fibrinogen/fibrin degradation product, VWF:RCo von Willebrand factor activity, TM thrombomodulin, FVIII coagulation factor VIII activity, Temperature body temperature, Fbg fibrinogen level, Cre creatinine concentration. Exact p values were obtained using the spearman's rank correlation test (two-sided) and shown in the figure. **b** Concentrations of platelet aggregates of patients with COVID-19 (39.1% of $n = 110$ biologically independent samples) and low D-dimer levels (≤1 μg/mL). **c** Comparison of the platelet aggregate concentrations of mild ($n = 19$ biologically independent samples), moderate ($n = 23$ biologically independent samples), and severe ($n = 1$ biologically independent samples) patients with COVID-19 and low D-dimer levels (≤1 μg/mL), measured on the highest concentration day of each hospitalized patient. $p = 2.82 \times 10^{-11}$ with the two-sided Kruskal–Wallis test. *P* values of every two adjacent classes were obtained using the Mann–Whitney *U*-test (two-sided) and shown in the figure. **d** ROC curve of the data shown in Fig. 5c. Source data are provided as a Source Data file.

discharging from the hospital or the patients staying at level 4 (requiring mechanical ventilation or ECMO), respectively.

## Discussion

The combined capability of massive image-based profiling, temporal monitoring, and big data analysis of circulating platelets and platelet aggregates in the blood of COVID-19 patients has offered previously

unattainable insights into the underlying process of COVID-19-associated microvascular thrombosis. Specifically, it has shown the strong link between the concentration of platelet aggregates and the severity and mortality of COVID-19. Furthermore, the positive correlation between the concentration of platelet aggregates and the vascular endothelial dysfunction characterized by the VWF:RCo, FVIII, and TM levels (Fig. 5a) is aligned with the currently accepted theory of the pathophysiological mechanism of COVID-19 in which

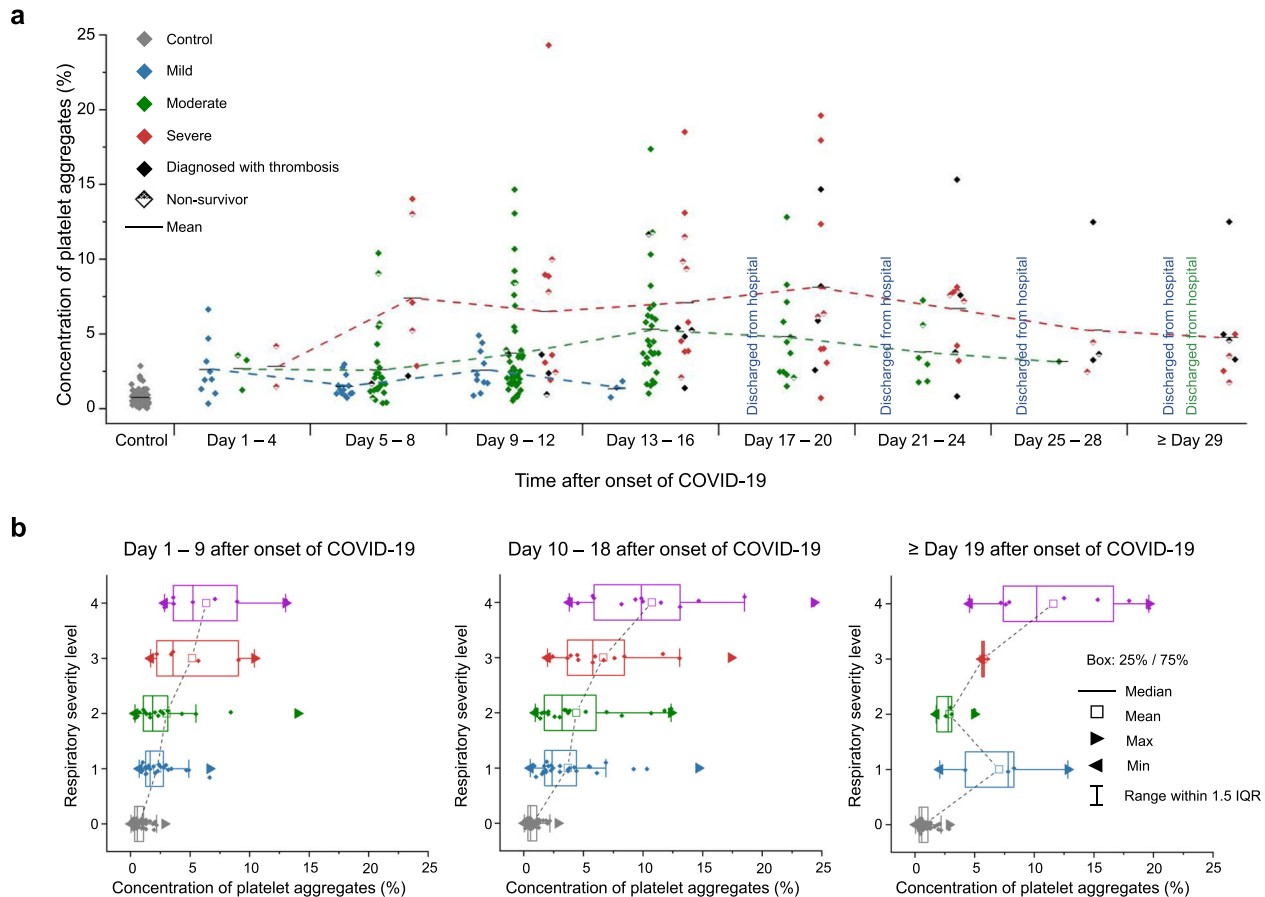

**Fig. 6 Temporal monitoring. a** Evolution of the platelet aggregate concentrations of different severity groups (mild, moderate, severe). Each patient's highest severity level during his/her hospitalization was used for the severity categorization and fixed during the entire hospitalization period to trace the evolution of the platelet aggregation concentration of each patient group. **b** Comparison of the respiratory severity level with the concentration of platelet aggregates ($n = 57$ biologically independent samples of Day 1–9 after the onset of COVID-19, $n = 78$ biologically independent samples of Day 10-18 after the onset of COVID-19, $n = 21$ biologically independent samples of Day $\geq$ 19 after the onset of COVID-19). The centerline indicates the median, box limits represent the first and third quartiles, and whiskers show all data within 1.5 times the interquartile range (IQR) of the lower and upper quartiles. Minima and maxima are shown with triangle marks. Level 0: control (gray). Level 1: without oxygen administration (blue); Level 2: with oxygen administration of 0.5–4 L/min (green); Level 3: with oxygen administration of $\geq$ 5 L/min (red); Level 4: with mechanical ventilation or ECMO (purple). Source data are provided as a Source Data file.

the viral entry into host cells is mediated by the ACE2 receptor, causing vascular endothelial damage and resulting in endothelial vascular dysfunction and vasculitis (i.e., the inflammation of blood vessels)[2,6]. Since VWF is a factor that mediates platelet adhesion and aggregation at the site of vascular endothelial injury, it is released into the blood in large amounts during the inflammation-mediated endothelial injury. Our findings (Figs. 3a and 5a) verified earlier reports on the correlation between the VWF level and the severity of COVID-19[51] by directly observing the increased concentration of platelet aggregates with its positive link to the severity of COVID-19. Additionally, our findings (Fig. 3a through Fig. 3c) also support a recently suggested mechanism in which platelets are hyperactive during COVID-19, not just via vasculitis, but also via the direct interaction of SARS-CoV-2 with platelets[47], which may account for the anomalous presence of excessive platelet aggregates found in nearly 90% of all COVID-19 patients in our study. Finally, based on the earlier reports that the lung is a major site of platelet biogenesis[52] and develops widespread microthrombi in severe COVID-19 patients[3], it can be speculated that the simultaneous activation of platelets and endothelial cells in the lung may be closely involved in the deterioration of respiratory conditions in COVID-19 patients and antiplatelet therapy may be effective for arterial oxygenation and improving clinical outcomes in severe COVID-19 patients[53].

Our findings suggest that measuring the concentration and distribution of circulating platelet aggregates is a potentially effective approach to evaluating the potential risk of micro-thrombus formation, a condition that can only be verified by postmortem examination. In fact, there are a number of autopsy reports showing that the primary cause of death in patients who died of COVID-19 pneumonia was respiratory failure due to diffuse alveolar damage with severe capillary congestion caused by microthrombi[10,18]. The strong correlation between the concentration of platelet aggregates and the oxygen administration severity and SpO$_2$ levels (Figs. 5a and 6a, b) suggests our method's ability to detect precursors to widespread microthrombus formation. Also, in our study, COVID-19 patients who were not diagnosed with thrombosis may have developed microthrombi which were too small to detect by medical imaging (e.g., CT and MRI), but were identified by our method as shown in Fig. 3a, c. In fact, a meta-analysis of COVID-19-associated thrombosis has reported that the incidence of venous thromboembolism differed significantly, depending on whether a screening test was performed[54]. It is possible that the incidence of thrombosis was missed in our clinical diagnosis, which may account for the mismatch between the severe cases and the confirmed onset of thrombosis in Fig. 6a. Finally, Fig. 5c indicates that our method is

sensitive to platelet activity as a potential precursor of micro-thrombus formation in COVID-19-associated microvascular thrombosis, while the D-dimer test is insensitive to it unless the severity of the patient's medical condition is high. Further studies are needed to directly verify the link between the concentration of platelet aggregates and the risk of microvascular thrombosis, for example, via comparing results from our method and autopsies.

The reason why excessive platelet aggregates were not found in the remaining 12.7% of the COVID-19 patients can be explained by recognizing two factors. First, the frequency of measurements on the mild and moderate patients was much lower than that of the severe patients, which led to a lower accuracy in finding the optimal timing for measuring the concentration of platelet aggregates (i.e., the highest concentration day) for the mild and moderate patients. In fact, the average numbers of measurements per patient were 1.9 (mild), 2.6 (moderate), and 7.3 (severe) times. The fact that the high AUC in the ROC curve of the concentration of platelet aggregates on the first measurement day is as large as 0.9 (as shown in Fig. 3e) indicates that both the sensitivity and specificity of our method are high despite the one-time measurements. This means that the low frequency of mea-surements is responsible for the lower concentration of platelet aggregates measured. Second, nearly all the COVID-19 patients received heparin-based anticoagulant treatment, which may have led to an overall variation in the concentration of platelet aggregates. Therefore, more frequent measurements without the influence of anticoagulant therapy (e.g., using animal models) are expected to improve the temporal resolution of image-based platelet aggregate profiling in assessing the varying severity of COVID-19 patients and hence improve the accuracy of the plots, trends, and correlations in Figs. 3a–c, 4a–c and 5b–c.

There are a few limitations to this retrospective observational study and potential solutions to them. First, all the subjects were hospitalized patients at the University of Tokyo Hospital alone, which may have introduced a selection bias. Further multi-hospital studies are needed to draw more general conclusions. Second, our method is not directly applicable to COVID-19 diagnosis because the identification of circulating platelet aggre-gates is not the direct evidence of microvascular thrombosis while extensive diffuse microthrombi are associated with multiorgan failure[1,3,7–11] and because it has not been conducted in patients under other pathological conditions. The relation between circu-lating platelet aggregates and microthrombi needs to be investi-gated via animal studies, in vivo optical microscopy, or high-resolution medical imaging to directly verify the link between the increased concentration of circulating platelet aggregates and the severity of microvascular thrombosis. Also, observation and analysis of circulating platelet aggregates in patients with other types of diseases (e.g., non-COVID-19 infectious diseases, sepsis, and microvascular thrombosis) need to be carried out. Moreover, a prospective study of prediagnosed patients with COVID-19 needs to be conducted to evaluate the diagnostic potential of our method. Third, a prospective study is needed to determine whe-ther the concentration of platelet aggregates can predict worsening respiratory status and the development of microthrombi as well as evaluate the efficacy of therapeutics. With these additional studies, our method may hold promise for predicting respiratory failure at an early stage and for developing treatment strategies for pre-venting the transition of patients to ventilatory management.

## Methods

**Human subjects.** This study was conducted with the approval of the Institutional Ethics Committee in the School of Medicine at the University of Tokyo [no. 11049, no. 11344] in compliance with the relevant guidelines and regulations. The subjects used in this study were patients who were clinically diagnosed with COVID-19 based on their reverse transcription-polymerase chain reaction (RT-PCR) test

results (Supplementary Data 1). Blood samples were collected as residual coagu-lation test samples (with 3.2% citrate) after the completion of requested clinical laboratory tests at the University of Tokyo Hospital. Blood cells in the samples were stored at room temperature while the residual plasma was cryopreserved at −80 °C. The negative control group was composed of 4 healthy subjects. Blood samples (with 3.2% citrate) from the healthy subjects were drawn multiple times on 67 different dates for preparing control samples. Likewise, to verify that the effect of using the residual coagulation test samples after the clinical laboratory tests on platelet aggregation was negligible, the positive control group was composed of seven hospitalized patients under no anticoagulant therapy and with no abnorm-ality confirmed by their coagulation tests (Supplementary Fig. 3), which indicated prothrombin time international normalized ratio (PT-INR), activated partial thromboplastin time (APTT), and fibrinogen levels of 0.88–1.10, 24–34 sec, and 1.68–3.55 g/L, respectively, and D-dimer levels of less than 1 μg/mL. Subjects under anticoagulant therapy were excluded. Clinical information (e.g., sex, age, severity, SpO$_2$ level, body temperature, respiratory severity) and laboratory test data were obtained from the electronic medical records of the patients using a standardized data collection form. Informed consent for participation in the study was obtained from the patients using an opt-out process on the webpage of the University of Tokyo Hospital. Patients who refused participation in our study were excluded. Written informed consent was obtained from the healthy subjects as well. The demographics, clinical characteristics, and laboratory findings of patients with COVID-19 are shown in Supplementary Data 1. The demographics, clinical characteristics, and laboratory findings of the positive control group are shown in Supplementary Data 5.

**Sample preparation.** Single platelets and platelet aggregates were enriched from whole blood by density-gradient centrifugation to maximize the efficiency of detecting platelets and platelet aggregates, as described in our previous report[26] with minor modifications. As shown in Fig. 2a, for analyzing the concentration of platelet aggregates, 500 μL of blood was diluted with 5 mL of saline. Platelets were isolated by using Lymphoprep (STEMCELLS, ST07851), a density-gradient med-ium, based on the protocol provided by the vendor. Specifically, the diluted blood was added to the Lymphoprep medium and centrifuged at 800 g for 20 min. After the centrifugation, 500 μL of the sample was taken from the mononuclear layer. Platelets were immunofluorescently labeled by adding 10 μL of anti-CD61-PE (Beckman Coulter, IM3605) and 5 μL of anti-CD45-PC7 (Beckman Coulter, IM3548) to the blood sample to ensure the detection of all platelets or platelet aggregates in the sample. Then, 500 μL of 2% paraformaldehyde (Wako, 163-20145) was added for fixation. Without the fixation, platelet aggregates in the sample would be dismantled as reported in our earlier work[26]. Therefore, the fixation process was performed at least within 4 h after the blood draw. The entire sample preparation was performed at room temperature.

**Clinical laboratory tests.** Blood samples from the patients were used for routine laboratory tests such as the leukocyte count (WBC), red cell count (RBC), platelet count (PLT), prothrombin time international normalized ratio (PT-INR), activated partial thromboplastin time (APTT), D-dimer level, fibrinogen level (Fbg), C-reactive protein concentration level (CRP), alanine transaminase concentration level (ALT), lactate dehydrogenase concentration level (LD), and creatinine kinase level (CK). The cryopreserved plasma was used for additional tests such as plasma plasmin-α$_2$-plasmin inhibitor complex level (PIC), fibrin monomer complex level (FMC), antithrombin level (AT), thrombin antithrombin complex level (TAT), von Willebrand factor activity level (VWF:RCo), fibrinogen/fibrin degradation product level (FDP), thrombomodulin level (TM), and coagulation factor VIII activity level (FVIII) (Fig. 5a, Supplementary Data 1, Supplementary Data 3). All the coagulation tests (i.e., AT, FDP, TAT, FMC, PIC, FVIII, VWF:RCo, TM, and D-dimer) were conducted on a CN6500 automatic coagulation analyzer (Sysmex, Japan). The D-dimer tests were performed with a latex-enhanced photometric immunoassay (LIAS AUTO D-dimer NEO, Sysmex, Japan). The laboratory reference range of the D-dimer test at the University of Tokyo Hospital was 0–1.0 μg/mL. Since the FVIII and VWF:RCo values of many blood samples exceeded the upper measurement limits of the analyzer (FVIII: 480%; VWF:RCo: 300%), their values were calculated based on the measured absorbance (FVIII: $y = 1090.5x^2 + 414.52x + 30.776$; VWF:RCo: $y = 1716x + 30.149$; where $x$ = absorbance, $y$ is expressed in units of %).

**Optical frequency-division-multiplexed microscope.** The FDM microscope is a high-speed, blur-free, bright-field imaging system based on a spatially distributed optical frequency comb as the optical source and a single-pixel photodetector as the image sensor. Since the optical frequency comb is composed of multiple beams which are spatially distributed, it is capable of simultaneously interrogating the one-dimensional spatial profile of a target object (e.g., a platelet, a platelet aggre-gate). In addition, since each discrete beam of the optical frequency comb is tagged by a different modulation frequency, a spatial-profile-encoded image can be retrieved by performing Fourier transformation on the time-domain waveform detected by the single-pixel detector. As shown in Supplementary Figure 1, we used a continuous-wave laser (Cobolt Calypso, 491 nm, 100 mW) as the laser source. Emitted light from the laser was split by a beam splitter, deflected and frequency-shifted by acousto-optic deflectors (Brimrose TED-150-100-488, 100-MHz

bandwidth), and recombined by another beam splitter. The resultant optical frequency comb was focused by an objective lens (Olympus UPLSAPO20X, NA:0.75) onto objects (e.g., single platelets, platelet aggregates) flowing at 1 m/s in a customized hydrodynamic-focusing microfluidic channel (Hamamatsu Photonics). Light transmitted through the flowing objects was collected by an avalanche photodiode (Thorlabs APD430A/M) and processed by a homemade LabVIEW program (LabVIEW 2016) to reconstruct the bright-field images. The line scan rate, spatial resolution, field of view, and number of pixels after making pixel aspect ratio corrections were 3 MHz, 0.8 μm, 53.6 μm x 53.6 μm, and 67 ×67 pixels, respectively. Fluorescence emitted from platelets labeled by anti-CD61-PE was also collected and used for triggering image acquisitions. Fluorescence emitted from leukocytes labeled by anti-CD45-PC7 was collected to identify platelet aggregates containing leukocytes. Image acquisition was performed at a high throughput of 100 - 300 events per sec (eps), where an event is defined as a single platelet or a platelet aggregate since red blood cells, leukocytes, and cell debris were not detected as events. The throughput was chosen to avoid clogging the microchannel, although the theoretical throughput of the machine was >10,000 eps.

**Statistical analysis**. The regions of objects (i.e., platelets, platelet aggregates) in bright-field images were segmented in MATLAB R2020a for calculating the concentration of platelet aggregates in each sample. First, a 10x interpolation by the interp2 function (MATLAB R2020a) was applied to each image for achieving segmentation results. Then, the outlines of the object regions were detected by using the edge detection function with the Canny method in MATLAB R2020a. Morphological operations like dilate, fill, and erode were applied to fill and refine the object regions for obtaining their masks as well as for eliminating the background noise. After the segmentation, the size of the object (i.e., a single platelet, a platelet aggregate) in each image was calculated by multiplying the pixel size of the segmented region, which was extracted by the regionprops function, and the pixel resolution after interpolation (80 nm x 80 nm/pixel). All the images with objective areas larger than 48 μm$^2$ were considered as the images of platelet aggregates in all the patient and healthy subject (control) samples. The concentration of platelet aggregates was defined by the ratio of the number of acquired images containing platelet aggregates to the total number of acquired images ($n = 25,000$) in each sample. In all 110 patient datasets, 106 of them have 25,000 images while only 4 of them (no. 6, 10, 12, 14) have 20,000 images due to a data-recording error, but this should not influence the statistical accuracy of our data analysis since the number of acquired images is significantly large. The presence of leukocytes in platelet aggregates was also identified by analyzing the morphology of the platelet aggregates by their images using custom codes in Python 3. For confirmation, the fluorescence signal of anti-CD61-PE and anti-CD45-PC7 was also used. CD61-PE/CD45-PC7 double-positive events were counted as platelet aggregates containing leukocytes. CD61-PE-positive and CD45-PC7-negative events that had CD61-PE signal intensity greater than a threshold value were counted as platelet aggregates excluding leukocytes. The presence of excessive platelet aggregates was determined by calculating the mean and standard deviation of the distribution of the concentration of platelet aggregates in the control samples and evaluating if the concentration of platelet aggregates in a patient sample exceeded the threshold (mean + standard deviation). If a tighter threshold (mean + two standard deviations) was used to calculate the presence of excessive platelet aggregates, then the ratio of the number of patients with excessive platelet aggregates to the number of all patients is 75.5%, while the ratio of the number of patients with excessive platelet aggregates and low D-dimer levels (≤1 μg/mL) to the number of all patients with low D-dimer levels (≤1 μg/mL) is 62.8%. Statistical analysis was performed using the rank-sum test including the Kruskal-Wallis test, Mann–Whitney U test, and Wilcoxon rank-sum test with significance levels at 0.05. Each statistical correlation value was obtained by calculating Spearman's rank correlation coefficient between the concentration of platelet aggregates on the highest concentration date and the corresponding clinical parameters. The line fitting function of the Origin 2021b software was used to draw linear fits to show the correlation between the concentration of platelet aggregates and each clinical parameter. A 95% confidence interval calculated from the standard error of measured y values is shown as a gray area in each panel (Supplementary Figs.4 and 5).

**Reporting Summary**. Further information on research design is available in the Nature Research Reporting Summary linked to this article.

## Data availability

The source data (Source Data 1) used in this study are available on the Zenodo database with access code 5592602 and are also available from the corresponding authors upon reasonable request. Source data are provided with this paper.

## Code availability

All the codes used in this study are available on the Zenodo database with access code 5592561 and are also available from the corresponding authors upon reasonable request.

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

## Acknowledgements
This work was supported by AMED JP20wm0325021 (M. N., K. G., Y. Y.), JSPS Core-to-Core Program (K. G.), JSPS KAKENHI grant numbers 19H05633 and 20H00317 (K. G.), ImPACT Program (CSTI, Cabinet Office, Government of Japan) (K. G.), White Rock Foundation (K. G.), Ogasawara Foundation (K. G.), Nakatani Foundation (K. G.), Konica Minolta Foundation (K. G.), and Charitable Trust Laboratory Medicine Research Foundation of Japan (M. N.). M. S., Ya. Z., X. Y., A. R. and G. R. were supported in part by National Institutes of Health, USA (award GM130825), and National Science Foundation, USA (award 1759802). We thank Kyoko Hasegawa and Yoshika Kusumoto for help with the sample preparation.

## Author contributions
K. G. conceived the work. H. K., Yu. Z., T. X. and K.G. designed the machine for massive image-based profiling of platelet aggregates. H. K. built the machine. M. N., J. T. and Yu. Z. prepared the blood samples. H. K., Y. I. and M. N. performed the image acquisition of platelets and platelet aggregates. Yu. Z., S. T., M. S., T. S. and G. R. analyzed the image data. J. H., S. T., Y. D., H. Z., K. H., R. K., W. P., M. S., Ya. Z. X. Y., A. R., G. R. and W. I. helped the image data acquisition and analysis. Yu. Z., M. N., T. X., N. N., S. M., Y. Y. and K. G. interpreted the analysis results. T. X., Y. Y. and K. G. supervised the work. All others participated in writing the paper.

## Competing interests
N. N. and K. G. are shareholders of CYBO, Inc. The other authors declare no competing interests.
