## [Peer Review File · Nature Communications]

Massive image-based single-cell profiling reveals high levels of circulating platelet aggregates in patients with COVID-19REVIEWER COMMENTS

Reviewer #1 (Remarks to the Author):

In the manuscript by Nishikawa et al entitled "The landscape of circulating Platelets aggregates in COVID19" the authors present a novel approach based on imaging cells in flow to address the highly relevant and important questions of platelet behaviours in different presentations of COVID19 with other clinical co-factors also considered. There is no question that this is a well-considered and well executed study that is addressing a very important questions in the presentation and pathogenesis of COVID19. I have now read the manuscript through several times and I simply cannot find a fault with either the technical aspects of the imaging flow cytometry, the sample preparation or the clinical meta-data integration and subsequent conclusions. If I was to make any slight criticism it would be that there are two platforms that could possibly do this in a more "easy access" setting that would be more amenable to a hospital testing lab or core technology platforms (Luminex ImageStream/FlowSight or the Thero Attune CytoPix). But this takes nothing away from the study as presented. We spend a lot of time in flow cytometry trying to avoid analysing doublets and aggregates, but this shows a beautiful example of when it is important to be able to not just identify, but also measure the magnitude of these aggregates as a direct correlate and possibly predictor of outcome in COVID19 disease. Due to the small size of platelets, the approach absolutely has to be image-based and could not be based on more classical approaches for identifying doublets and aggregates by non-imaging flow cytometry (pulse area/height/width).

It is my recommendation that this work be accepted for rapid publication and I congratulate the authors on a very powerful piece of work.

Reviewer #2 (Remarks to the Author):

This study represents a significant body of work; the authors undertook a large number of evaluations for each patient and show some very interesting results. The methodology for the investigation of the platelet aggregates is not widely used, so the applicability of the results, as a means to predict severity of COVID-19 is limited. Nonetheless, a nice study to show that platelet aggregates, not otherwise easily identified in patients, correlates with disease severity. I don't have any suggestions for the authors.

Reviewer #3 (Remarks to the Author):

This is an interesting technical advance in monitoring platelet aggregates in circulating blood. Essentially, in simple terms, the authors' are looking at cells flowing past a detector and measuring the number of single platelets, platelet aggregates, or platelet leukocyte aggregates. They can then do a correlation between the distribution of these populations and disease severity, and show a correlation with overlap between groups. This is a more sophisticated way than other assays of platelet activation e.g. platelet-leukocyte aggregates, but the question is whether this is an advance in disease diagnosis. A prospective study needs to be performed to answer this.

The microscope is custom built system which acts like a sophisticated flow cytometer (apologies to the authors if they do not like this analogy). It monitors the fluorescence signal and when a platelet is detected (via CD61 signal) it triggers capturing of a brightfield image. In this way there is no need to collect images of every cell and they discard the ones that are not platelets I don't understand fully the physics of how they achieve this, but there are a few publications out there on the principle and development of the systems, including commercial systems such as the image stream?

I see this as a very nice advance but with unproven practical significance for clinical management.

To Reviewer #1:

We are grateful to the Reviewer for taking the time to review our manuscript and give us his/her valuable comments. We have taken all the comments into consideration and have made appropriate changes to the manuscript. Our point-by-point response appears below, in which we first echo the Reviewer's comments (shown in italic) and then respond to them. All the changes to the manuscript and Supplementary Information are highlighted in red.

Reviewer #1's comment #1:

In the manuscript by Nishikawa et al entitled "The landscape of circulating Platelets aggregates in COVID19" the authors present a novel approach based on imaging cells in flow to address the highly relevant and important questions of platelet behaviours in different presentations of COVID19 with other clinical co-factors also considered. There is no question that this is a well-considered and well executed study that is addressing a very important questions in the presentation and pathogenesis of COVID19. I have now read the manuscript through several times and I simply cannot find a fault with either the technical aspects of the imaging flow cytometry, the sample preparation or the clinical meta-data integration and subsequent conclusions.

Authors' response:

We thank the Reviewer for giving us the positive comment and reading our manuscript several times.

Reviewer #1's comment #2:

If I was to make any slight criticism it would be that the there are two platforms that could possibly do this in a more "easy access" setting that would be more amenable to a hospital testing lab or core technology platforms (Luminex ImageStream/FlowSight or the Thero Attune CytoPix). But this takes nothing away from the study as presented.

Authors' response:

We thank the Reviewer for the comment. We agree that the commercial systems can provide easier access in clinical settings than our method, but our method is advantageous in the throughput of detecting all platelet events (e.g., single platelets, platelet-platelet aggregates, and platelet-leukocyte aggregates) than previous imaging flow cytometers. Specifically, our microscope's image acquisition of all platelet events (e.g., single platelets, platelet-platelet aggregates, and platelet-leukocyte aggregates) was triggered by detecting fluorescence signals from anti-CD61-PE-labeled platelets, which is advantageous over previous imaging flow cytometers in that our method can only focus on platelet events while avoiding its throughput from being consumed by non-platelet events. To clarify this point, we have added the following text to the second paragraph in the Introduction section: "The FDM microscope's image acquisition of all platelet events (e.g., single platelets, platelet-platelet aggregates, and platelet-leukocyte aggregates) was triggered by detecting fluorescence signals from anti-CD61-PE-labeled platelets (see "optical frequency-division-multiplexed microscope" in the Methods section for details), which is advantageous over previous imaging flow cytometers in that our method can only focus on platelet events while avoiding its throughput from being consumed by non-platelet events."

Reviewer #1's comment #3:

We spend a lot of time in flow cytometry trying to avoid analysing doublets and aggregates, but this shows a beautiful example of when it is important to be able to not just identify, but also measure the magnitude of these aggregates as a direct correlate and possibly predictor of outcome in COVID19 disease. Due to the small size of platelets, the approach absolutely has to be image-based and could not be based on more classical approaches for identifying doublets and aggregates by non-imaging flow cytometry (pulse area/height/width). It is my recommendation that this work be accepted for rapid publication and I congratulate the authors on a very powerful piece of work.

Authors' response:

We thank the Reviewer for recognizing our work and recommending rapid publication.

To Reviewer #2:

We are grateful to the Reviewer for taking the time to review our manuscript and give us his/her valuable comments. All the changes to the manuscript and Supplementary Information in relation to our response to the other two reviewers are highlighted in red.

Reviewer #2's comment #1:

This study represents a significant body of work; the authors undertook a large number of evaluations for each patient and show some very interesting results. The methodology for the investigation of the platelet aggregates is not widely used, so the applicability of the results, as a means to predict severity of COVID-19 is limited. Nonetheless, a nice study to show that platelet aggregates, not otherwise easily identified in patients, correlates with disease severity. I don't have any suggestions for the authors.

Authors' response:

We thank the Reviewer for recognizing our work and giving the positive comment.

To Reviewer #3:

We are grateful to the Reviewer for taking the time to review our manuscript and give us his/her valuable comments. We have taken all the comments into consideration and have made appropriate changes to the manuscript. Our point-by-point response appears below, in which we first echo the Reviewer's comments (shown in italic) and then respond to them. All the changes to the manuscript and Supplementary Information are highlighted in red.

Reviewer #3's comment #1:

This is an interesting technical advance in monitoring platelet aggregates in circulating blood. Essentially, in simple terms, the authors' are looking at cells flowing past a detector and measuring the number of single platelets, platelet aggregates, or platelet leukocyte aggregates. They can then do a correlation between the distribution of these populations and disease severity, and show a correlation with overlap between groups. This is a more sophisticated way than other assays of platelet activation e.g. platelet-leukocyte aggregates, but the question is whether this is an advance in disease diagnosis. A prospective study needs to be performed to answer this.

Authors' response:

We thank the Reviewer for the positive comment and recognition of our work. Since this work is a retrospective observational study of COVID-19-associated microvascular thrombosis, we agree with him/her that a prospective study is needed to determine whether it can be used in clinical diagnosis. To clarify this point, we have revised the last paragraph in the Discussion section by providing additional text about the prospective study: "There are a few limitations to this retrospective observational study and potential solutions to them. First, all the subjects were hospitalized patients at the University of Tokyo Hospital alone, which may have introduced a selection bias. Further multi-hospital studies are needed to draw more general conclusions. Second, our method is not directly applicable to COVID-19 diagnosis because the identification of circulating platelet aggregates is not the direct evidence of microvascular thrombosis while extensive diffuse microthrombi are associated with multiorgan failure and because it has not been conducted in patients under other pathological conditions. The relation between circulating platelet aggregates and microthrombi needs to be investigated via animal studies, in vivo optical microscopy, or high-resolution medical imaging to directly verify the link between the increased concentration of circulating platelet aggregates and the severity of microvascular thrombosis. Also, observation and analysis of circulating platelet aggregates in patients with other types of diseases (e.g., non-COVID-19 infectious diseases, sepsis, and microvascular thrombosis) need to be carried out. Moreover, a prospective study of pre-diagnosed patients with COVID-19 needs to be conducted to evaluate the diagnostic potential of our method. Third, a prospective study is needed to determine whether the concentration of platelet aggregates can predict worsening respiratory status and the development of microthrombi as well as evaluate the efficacy of therapeutics. With these additional studies, our method may hold promise for predicting respiratory failure at an early stage and for developing treatment strategies for preventing the transition of patients to ventilatory management."

Reviewer #3's comment #2:

The microscope is custom built system which acts like a sophisticated flow cytometer (apologies to the authors if they do not like this analogy). It monitors the fluorescence signal and when a platelet is detected (via CD61 signal) it triggers capturing of a brightfield image. In this way there is no need to collect images of every cell and they discard the ones that are not platelets I don't understand fully the physics of how they achieve this,

but there are a few publications out there on the principle and development of the systems, including commercial systems such as the image stream?

Authors' response:

We thank the Reviewer for the comment. Our method is advantageous in the throughput of detecting all platelet events (e.g., single platelets, platelet-platelet aggregates, and platelet-leukocyte aggregates) than previous imaging flow cytometers. Specifically, our microscope's image acquisition of all platelet events (e.g., single platelets, platelet-platelet aggregates, and platelet-leukocyte aggregates) was triggered by detecting fluorescence signals from anti-CD61-PE-labeled platelets, which is advantageous over previous imaging flow cytometers in that our method can only focus on platelet events while avoiding its throughput from being consumed by non-platelet events. To clarify this point, we have added the following text to the second paragraph in the Introduction section: "The FDM microscope's image acquisition of all platelet events (e.g., single platelets, platelet-platelet aggregates, and platelet-leukocyte aggregates) was triggered by detecting fluorescence signals from anti-CD61-PE-labeled platelets (see "optical frequency-division-multiplexed microscope" in the Methods section for details), which is advantageous over previous imaging flow cytometers in that our method can only focus on platelet events while avoiding its throughput from being consumed by non-platelet events."

Reviewer #3's comment #3:

I see this as a very nice advance but with unproven practical significance for clinical management.

Authors' response:

We thank the Reviewer for the comment. We ask him/her to refer to our response to his/her first comment.

REVIEWERS' COMMENTS

Reviewer #3 (Remarks to the Author):

The authors have included a section on the limitations of this study and the need for a prospective study that addresses my major concern.

Reviewer #3's comment:

The authors have included a section on the limitations of this study and the need for a prospective study that addresses my major concern.

Authors' response:

We thank the reviewer for the positive comment.